# Topical gabapentin solution for the management of burning mouth syndrome: A retrospective study

Amanda Gramacy[1☯], Alessandro Villa[1,2☯]*

1 Department of Orofacial Sciences, School of Dentistry, University of California San Francisco, San Francisco, California, United States of America, 2 Oral Medicine, Oral Oncology and Dentistry, Miami Cancer Institute, Baptist Health South Florida, Miami, Florida, United States of America

☯ These authors contributed equally to this work.
* Alessandro.villa@ucsf.edu

## Abstract

### Objective

The aim of this retrospective study was to evaluate the effectiveness and safety of topical gabapentin solution (250 mg/mL) for the management of burning mouth syndrome (BMS).

### Study design

A retrospective chart review was conducted of all patients diagnosed with BMS and managed with gabapentin 250 mg/mL solution (swish and spit) between January 2021 and October 2022. Patient-reported outcomes included changes in burning score ranked on a 10-point numeric rating scale (NRS) and reported adverse drug reactions (ADR). Wilcoxon signed-rank test was used to assess differences in the oral burning score ranked on a NRS (0–10) between the baseline visit and the second visit.

### Results

A total of 19 patients (68.4% females) with BMS were included and evaluated for follow-up at a median of 86 days (range: 29–195). Overall, patients reported a median 2-point burning decrease on a 0–10 NRS between the baseline visit and the second visit (p < 0.01). ADRs were reported by 3 patients (15.8%).

### Conclusion

Although this was a small retrospective study, BMS management with topical gabapentin (250 mg/mL) appears to be effective and well-tolerated. Future randomized prospective studies are needed to verify these preliminary findings.

**Data Availability Statement:** All relevant data are within the paper and its Supporting information files.

**Funding:** The authors received no specific funding for this work.

**Competing interests:** The authors have declared that no competing interests exist.

## Introduction

Burning Mouth Syndrome (BMS) is a chronic pain condition characterized by the presence of a burning sensation/pain of the oral cavity without any clinically evident signs of lesions or systemic causes [1]. Prevalence rates of BMS in the general population range between 0.7% to 15%, with higher rates seen in older females [2]. The exact etiology of BMS is poorly understood although there is a predisposition to peri-/postmenopausal women [2]. The primary symptoms include oral burning although patients may present with other sensory symptoms including xerostomia and dysgeusia [3]. Oral burning can occur in more than one oral site of the oral cavity with the most common areas affected being the anterior two thirds of the tongue, the anterior hard palatal mucosa, and the lower lip mucosa [4].

The pathophysiology of BMS is poorly understood and may be related to both physiological and psychological components [5]. Evidence suggests that the mechanism of BMS is neuropathic in nature [5]. In the neuropathic pathophysiology theory, sensory dysfunction is associated with small and/or large fiber neuropathy where there is axonal degeneration of epithelial and subpapillary nerve fibers in the affected epithelium of the oral mucosa [6]. There is also an abnormal interaction between the sensory functions of facial and trigeminal nerves. BMS is considered a nociplastic pain disorder in that the pain arises from altered nociception despite no evidence of tissue damage causing the activation of peripheral nociceptors and no evidence of a lesion causing the pain [7]. Although there is no universally accepted diagnostic criteria for BMS, the diagnosis of BMS is a diagnosis of exclusion that can be determined through analysis of symptoms, medical history, and physical and laboratory testing [1, 8].

Currently, there is no definitive treatment or cure for BMS, and treatments have largely focused on symptoms relief [9]. Treatment response of BMS has proven to be highly variable, likely due to both the multifactorial and neuropathic nature of the condition [1]. Systemic medications for the management of BMS includes tricyclic antidepressants, anticonvulsants, benzodiazepines, and opioids have shown variable response rates and a risk of short-term side effects [1]. Several topical treatments have been also proposed and include capsaicin, compounded clonazepam solution as a swish and spit, lidocaine, and benzydamine hydrochloride [1]. However, systematic reviews of topical interventions for the management of BMS show there is a current lack of strong evidence to support topical therapy [10, 11]. Topical interventions for management of BMS are usually effective, but the quality of evidence remains low [10, 11].

Systemic gabapentin has been found to be beneficial in relieving burning symptoms in patients with BMS [12]. However, systemic gabapentin has been associated with various side effects and a potential for misuse and overdose [13, 14]. Given these potential side effects, there is considerable interest in identifying safe and effective topical therapies for the management of BMS. In our work, we hypothesized that patients with BMS may show an improvement of BMS related symptoms using a gabapentin solution (250 mg/ 5mL) as a swish and spit. As such, the objective of this retrospective study was to evaluate the effectiveness and safety of topical gabapentin solution (250 mg/5mL) for the management of BMS.

## Materials and methods

### Patient identification

We conducted a retrospective electronic medical chart review and identified all patients who were prescribed topical gabapentin solution (250mg/5mL) at the University of California San Francisco (UCSF) Sol Silverman Oral Medicine clinic between January 2021 and October 2022. Chart reviews were accessed for research purposes between 7/1/2022 and 1/30/2023.

Patients were diagnosed with BMS by oral medicine specialists based on The International Classification of Headache Disorders classification [15]. Only patients with a diagnosis of BMS and with at least one follow-up visit were included in this analysis. All patients prescribed with the topical gabapentin solution were instructed to swish and spit 5mL of the solution for 5 minutes without swallowing two to four times a day. Patients that were prescribed topical gabapentin for reasons other than BMS were excluded from the study (Fig 1). The study was approved by the UCSF Institutional Review Board. Consent was waived given the retrospective nature of this study.

## Patients' characteristics

We extracted data from the electronic medical records and included patient demographics, past medical history, smoking and alcohol consumption, comorbidities, type of oral dysesthesia(s), intensity of burning/sensitivity as measured on a 0-10-point numeric rating scale (NRS; with 0 being no pain and 10 being very intense pain at each visit), oral symptoms distribution in the oral cavity, current psychiatric medications, concomitant and past treatments for BMS, and adverse drug reactions (ADR).

BMS was classified into three types: Type 1, with burning sensation developing in the late morning, gradually increasing in severity during the day, and reaching its peak intensity by evening; Type 2, with continuous symptoms throughout the day and difficulty getting to sleep at night; and Type 3, with pain-free periods during the day [16]. We included patients that had both primary and secondary BMS. Patients with an underlying medical condition were stable and have had these medical conditions for many years.

## Statistical analyses

Descriptive statistics were used to summarize patients' characteristics, burning scores and ADR. Treatment response was assessed using patient-reported outcome measures. The Wilcoxon signed-rank test was used to assess differences in the oral burning score ranked on a NRS (0–10) between the baseline visit and the second visit. We calculated the median pain score at the baseline visit and at the second visit and compared the two values using the Wilcoxon signed rank test (Table 1). A p value $< 0.05$ was considered statistically significant. The p value is a p for trend.

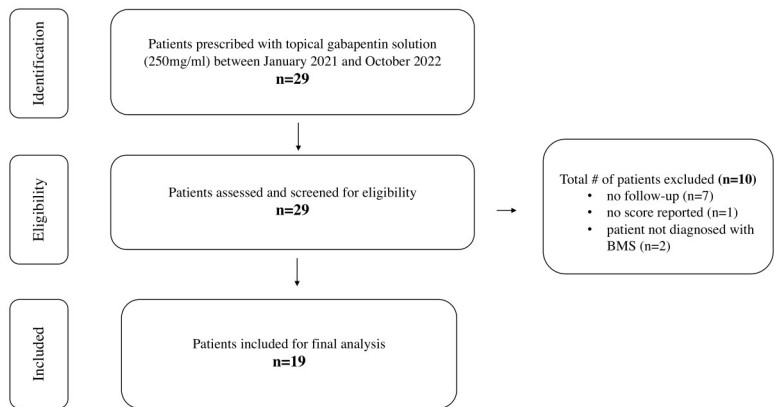

**Fig 1. Identification and eligibility of BMS patients.**

**Table 1. Burning NRS* score.**

| Median NRS Score at Baseline Visit | Median NRS Score at Second Visit | p value |
|---|---|---|
| 5 | 3 | p< 0.01 |

*NRS = Numerical Rating Scale

## Results

A total of 29 patients were prescribed gabapentin (250 mg/5mL) solution between January 2021 and October 2022. A total of 10 patients were excluded due to missing numbers or loss to follow-up. A total of 19 BMS patients met the inclusion criteria and were included in the final analysis (Fig 1).

Most patients were females (n = 13; 68.4%) with a mean age of 68 years and a standard deviation of 11 (range: 46–85) (Table 2). The majority of patients were non-smokers (73.7%) and consumed alcoholic beverages (57.9%). Patients reported taking concomitant agents, with the most common being over the counter moisturizers (89.5%), followed by antidepressants (42.1%) and alpha lipoic acid (31.6%). All patients (100%) reported having at least one comorbidity, with the most common being depression (42.1%) (for a full list of comorbidities see Table 2).

The median burning NRS score at first visit was 5 (range 1 to 10; 95% confidence interval [CI]: 4.5–7.3) and at second visit was 3 (range 0 to 8; 95% CI: 2.1–4.4) with a 2-point median reduction in the burning NRS score ($p < 0.01$) in a median of 86 days (range: 29–195) (Table 3). Five patients returned for a third visit (median score: 4; range 2–7; 95% CI: 1.8–7.5). The clinical pattern of BMS varied between patients, with 21.1% exhibiting Type I, 21.1% exhibiting Type II, 36.8% exhibiting Type III and 21% exhibiting a BMS pattern not defined. Most patients reported at least one other oral symptom aside from oral burning including xerostomia (n = 14; 76.2%) and dysgeusia (n = 8; 42.9%). The most affected oral cavity sites were the lower lip mucosa (42.1%), tip of tongue (31.6%), and upper lip mucosa (26.3%).

ADRs of topical gabapentin occurred in three patients (15.8%). All three patients (15.8%) discontinued the topical gabapentin solution after the second visit due to dry mouth, sedation and the solution being uncomfortable to keep in the mouth (lip sensitivity), respectively (Table 4). No patients reported any tingling sensation or "bad taste" with the gabapentin 250mg/mL solution.

## Discussion

BMS has proven to be challenging to manage due to its complex and poorly understood pathogenesis [17]. Systemic therapy is with antidepressants, anticonvulsants, benzodiazepines, and for severe cases with opioids. These pharmacologic therapies carry a risk of short term-side effects such as fatigue, dizziness, and disorientation, as well as serious long-term side effects such as physical addiction, and dementia [13, 14]. The identification of effective topical therapies for managing BMS has generated significant interest due to the potential side effects associated with existing options.

Systemic gabapentin is considered a well-tolerated anticonvulsant drug with a favorable pharmokinetic profile and broad therapeutic index which has been used over the years as a systemic option for BMS with mixed results [18–20]. However, systemic gabapentin has been associated with adverse effects that may cause dizziness, drowsiness and confusion, especially in older adults [21]. Given these adverse effects from systemic therapy, we assessed the safety and effectiveness of a topical gabapentin solution (250mg/mL) for the management of BMS.

**Table 2. Patient characteristics at baseline.**

| | n (%) |
|---|---|
| **Sex at birth** | |
| Male | 6 (31.6) |
| Female | 13 (68.4) |
| **Tobacco use** | |
| Current | 1 (5.3) |
| Former | 3 (15.8) |
| Never | 14 (73.7) |
| Did not specify | 1 (5.3) |
| **Alcohol Consumption** | |
| Current | 11 (57.9) |
| Former | 0 (0.0) |
| Never | 5 (26.3) |
| Did not specify | 3 (15.8) |
| **Concomitant agents** | **Current** |
| Over the counter moisturizers* | 17 (89.5) |
| Antidepressants | 8 (42.1) |
| Alpha lipoic acid | 6 (31.6) |
| Viscous lidocaine | 4 (21.1) |
| Anticonvulsants/Anxiolytics (systemic) | 3 (15.8) |
| **Comorbidities** | **Current** |
| Depression | 8 (42.1) |
| Hypercholesteremia | 7 (36.8) |
| Anxiety | 5 (25.3) |
| Hypothyroidism | 4 (21.1) |
| Heart conditions | 4 (21.1) |
| Hypertension | 3 (15.8) |
| GERD | 3 (15.8) |
| Breast cancer | 3 (15.8) |
| Arthritis | 2 (10.5) |
| Insomnia | 2 (10.5) |
| Diabetes mellitus type II | 1 (5.3) |
| Osteoporosis | 1 (5.3) |
| Post-Traumatic Stress Disorder (PTSD) | 1 (5.3) |

* Includes dry mouth lozenge and saliva substitute products (such as Biotene® and Xylimelts®)

When the treatment response was considered, patients reported a median 2-point burning decrease on the 0–10 NRS between the baseline visit and the second visit (p < 0.01). Of note, during the third visit there was a median 1-point burning increase (range 1–7) on the 0–10 NRS between the second and third visit. Of the 19 patients included in the study, only 5 patients completed or returned for their third visit. Thus, the NRS score on the third visit was not fully representative of all patients included in the study.

The precise mechanism of action of gabapentin remains unclear as to how it exerts its therapeutic effect. Systemic gabapentin inhibits the action of α2δ-1 subunits, thus decreasing the density of pre-synaptic voltage-gated calcium channels and subsequent release of excitatory neurotransmitters [7]. A topical gabapentin approach has shown to have a positive analgesic

**Table 3. Burning characteristics.**

| Burning Intensity Score (0–10 NRS) | Median (range) | Mean and standard deviation |
|---|---|---|
| Baseline Visit | 5 (2.5–10) | x̄ = 5.9 |
| | | σ = 2.80 |
| Second Visit | 3 (0–8) | x̄ = 3.0 |
| | | σ = 2.33) |
| Third Visit* | 4 (1–7) | x̄ = 4.6 |
| | | σ = 2.30 |
| Median # of days between Baseline and second visit | 86 (29–195) | |
| **Clinical Pattern of BMS** | **n (%)** | |
| Type 1 | 4 (21.1) | |
| Type 2 | 4 (21.1) | |
| Type 3 | 7 (36.8) | |
| Pattern not defined | 4 (21.0) | |
| **Other oral symptoms** | **n (%)** | |
| Xerostomia | 14 (76.2) | |
| Dysgeusia | 8 (42.9) | |
| Hypogeusia | 2 (14.3) | |
| Roughness | 3 (15.8) | |
| Numbness/tingling | 3 (15.8) | |
| Other** | 4 (21.1) | |

Note: some patients have not been seen yet for their third visit

*Data for the third visit was available for only 5 patients

*Data for the third visit was available for only 5 patients

**Other oral symptoms included: gritty/strange saliva, swelling, throat constriction

Abbreviations: BMS: burning mouth syndrome

effect with fewer side effects in other neuropathic pain syndromes such as severe postherpetic neuralgia and chronic sciatic nerve constriction injury [22, 23]. We can hypothesize that the mechanism of action of topical gabapentin is similar to systemic gabapentin by blocking the $\alpha2\delta$-1 subunits present in nociceptive neurons, but that the analgesic effect is produced locally. Nociceptors in mucosal barriers contain various types of receptors that bind different ligands which influence the generation of pain transmitting action potentials. Topical gabapentin could traverse tissue and increase the nociceptive threshold by stabilizing the membranes of specific nociceptors [24].

The topical pharmacological approach for the management of BMS was initially introduced by Gremeau-Richard et al [25]. In their study they evaluated the efficacy of topical clonazepam

**Table 4. Adverse drug reactions (ADR).**

| Adverse Drug Reactions | n (%) |
|---|---|
| Discontinued due to adverse effects | 3 (15.8) |
| Sedation | 1 (5.3) |
| Tingling | 0 (0.0) |
| "Bad" taste | 0 (0.0) |
| Dry mouth | 1 (5.3) |
| Uncomfortable to keep in mouth | 1 (5.3) |

in 48 patients affected by BMS. Patients were instructed to dissolve a 1.0 mg tablet of either clonazepam or placebo in the mouth for three minutes, and spit out without swallowing, three times daily for 14 days. Pain intensity was measured on a 11-point numerical scale and showed a statistically significant decrease of pain score (2.5 +/- 0.6) in the treatment group compared with the placebo group (0.6 +/- 0.4) with two-thirds of patients reporting a significant improvement. Topical clonazepam use was not associated with any significant adverse effects like those associated with systemic antidepressants or antianxioltyics [25]. In a separate study, Kuten-Schorrer *et al.* evaluated and compared the effectiveness of two concentrations of topical clonazepam solution (0.1 mg/mL and 0.5 mg/mL) for the management of oral dysesthesia (OD) in 57 patients [9]. Of the 32 patients in the 0.1-mg/mL cohort, 13 patients (41%) reported an improvement of at least 50%, compared to the 25 patients in the 0.5-mg/mL cohort where 23 patients (92%) reported a symptomatic improvement of at least 50% [9]. Similar to our study, the ADR were reported in 9 out of 58 patients (15.5% vs 15.8%) [9]. Our study examining the use of topical gabapentin for treatment of BMS yielded similar symptomatic improvements to those noted in both aforementioned studies evaluating topical clonazepam, making it a possible alternative.

The three types of ADRs recorded in our study were sedation, "uncomfortable to keep in mouth" and xerostomia, which are minor in nature. No serious ADRs (defined as a reaction that may result in death, is life threatening, requires hospitalization or prolongation of current hospital stay, or causes persistent or significant disability) were reported in our study. In comparison, in a different retrospective study which analyzed the safety and tolerability of treatment with topical clonazepam solutions (0.1 mg/mL, 0.5 mg/mL) for management of OD in 162 patients over a follow-up period of 6 weeks, there were 38 patients (23%) with ADRs [26]. Kuten-Schorrer *et al.* found seven different types of ADRs: sedation (62%), altered mental status (7%), dizziness (7%), burning increase, nausea, skin reaction and other [26]. Of note, two patients in the Kuten-Schorrer *et al.* study were involved in motor vehicle accidents where both patients attributed their accident to a state of sedation secondary to use of topical clonazapem [26]. The total prevalence of ADRs in our study (15.8%) were comparatively lower than what was observed by Kuten-Schorrer *et al.* in their retrospective study for patients receiving topical clonazepam solution for treatment of OD (23%) and by what was observed by Gremeau-Richard *et al.* in their randomized controlled trial evaluating topical clonazepam for the management of stomatodynia and BMS (37%) [25, 26].

Systemic gabapentin is an antiepileptic and anxiolytic agent and works by inhibiting calcium influx and subsequent release of excitatory neurotransmitters and is absorbed slowly after oral administration to achieve an analgesic effect [27]. Topical gabapentin used as a swish-and-spit solution target the underlying mechanisms of peripheral sensitization and pain leading to an analgesic effect. The exact mechanism of analgesic action of topical gabapentin is unclear but is thought to influence voltage-gated calcium channels, N-methyl-D-aspartate receptor receptors, potassium channels, and inflammatory mediators, leading to reduced neuronal hyperexcitability and antinociception [28]. It is important to consider that topical treatments including topical clonazepam and topical gabapentin for management of BMS may inadvertently have some sort of systemic effect as patients may swallow or ingest remnants of the solution or dissolved tablet.

Two of the most common comorbidities exhibited by BMS patients in our study were depression (n = 8; 42.1%) and anxiety (n = 5; 25.3%). An additional 8 patients (42.1%) also reported having a history of stress. BMS is commonly associated with mood and psychiatric disorders, with generalized anxiety disorder and depressive disorder being two of the most common [29, 30]. A previous study looking at the association between depression and BMS found that patients with depression were 3.08 times more likely to develop BMS than patients

without depression, and female patients with depression were even more prone to new-onset BMS (3.87 times higher) [31].

The present study has several limitations that should be addressed due to its retrospective nature. First, there was little to no documentation of compliance to the prescribed regimen of gabapentin (250mg/5mL) solution. Even though patients were instructed to rinse 2–4 times a day, we could not record the exact frequency of the rinses. Second, the follow-up period was variable, ranging from 29 days to 195 days making it challenging to compare patients' response to treatment at similar follow-up intervals. Additionally, many patients had to be excluded from our final analysis because some providers failed to report a baseline or follow-up burn score. Our study also exhibited clinical heterogeneity within our patient pool with respect to patient characteristics and thus it is difficult to generalize our findings to all patients with BMS. The study also exhibited a relatively small sample size, mostly due to missing date or loss to follow up. However, our work is also characterized by several strengths. Efforts to collect ADR data at the second and third visits minimized the risk of attrition bias as patients that have an ADR are less likely to report back for a follow-up compared with patients that may be responding well to the prescribed regimen at second and third visits. Additionally, utilizing a NRS score to evaluate effectiveness of topical gabapentin solution allowed for this study to be easily comparable to other studies that used similar methods to analyze responses to treatment for BMS.

In conclusion, this small retrospective study showed that gabapentin (250mg/5mL) solution was a safe and well-tolerated treatment in minimizing burning and pain in BMS patients. Topical approach to gabapentin may be considered by clinicians as an alternative to systemic gabapentin for the treatment of BMS with minimal ADRs. Looking forward, prospective, randomized and placebo-controlled studies should be used to confirm these initial results. Nevertheless, our findings can be an important reference for oral healthcare providers to refer to as a means of managing BMS.

## Supporting information

**S1 Dataset.**
(XLSX)

## Author Contributions

**Data curation:** Amanda Gramacy.

**Formal analysis:** Amanda Gramacy.

**Investigation:** Amanda Gramacy.

**Methodology:** Amanda Gramacy.

**Writing – original draft:** Amanda Gramacy.

**Writing – review & editing:** Alessandro Villa.

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
