## [Decision Letter · Decision Letter 0]

3 Oct 2023

PONE-D-23-22592Topical gabapentin solution for the management of Burning Mouth Syndrome: a retrospective studyPLOS ONE

Dear Dr. Villa,

Thank you for submitting your manuscript to PLOS ONE. After careful consideration, we feel that it has merit but does not fully meet PLOS ONE’s publication criteria as it currently stands. Therefore, we invite you to submit a revised version of the manuscript that addresses the points raised during the review process.

We look forward to receiving your revised manuscript.

Kind regards,

Claudia Sommer

Academic Editor

PLOS ONE

Journal Requirements

Reviewers' comments:

Reviewer's Responses to Questions

**Comments to the Author**

1. Is the manuscript technically sound, and do the data support the conclusions?

Reviewer #1: Yes

Reviewer #2: Yes

2. Has the statistical analysis been performed appropriately and rigorously? 

Reviewer #1: Yes

Reviewer #2: Yes

3. Have the authors made all data underlying the findings in their manuscript fully available?

Reviewer #1: Yes

Reviewer #2: Yes

4. Is the manuscript presented in an intelligible fashion and written in standard English?

Reviewer #1: Yes

Reviewer #2: Yes

5. Review Comments to the Author

Reviewer #1: The Authors presented an interesting clinical restrospective study on topical solution of gabapentin for BMS. The subject is clinicaly important due to limited effectivness of available treatment modalities.

The manuscript needs some minore recvisions before publication.

Line 59-60 - propose to decribe the pathophysiology of BMS more precisely (alternatively in discussion) , including some molecular and cellular mechanisms contributing to neuropathic component of BMS, what is important regarding the mechanism of action of gabapentin in this clinical entity. In the pathophysiology of BMS , the nociplastic component may be involved, propose to add .

section Discussion :

propose to discuss more precisely the molecular and cellular mechanism of action of gabapentin , particularly in terms of topical administration. Which mechanisms of gabapentin MOC might be involved in BMS treatment , which types of cells may be targeted by gabapentin? Propose to refer as well to clinical studies presenting other pain syndromes treated with topical gabapentin, including mucosistis if there are studies avaialble.

Despite small group of patients studied , the study is relevant for clinical purposes and after minor revisions may be published

Reviewer #2: - Introduction

1) On line 55, I suggest removing the sentence: “tingling or changes in salivary gland function”.

- Materials and Methods

1) I suggest adding to your methodology the inclusion of patients with primary and secondary BMS, since it was mentioned that all patients in the sample had comorbidities such as type II diabetes, hypothyroidism,...

- Results

1)I think there is no need to subdivide the results by topic.

2) I suggest removing the information regarding the patients' age from table 1, since it has already been mentioned in the text. It is preferable to use the mean and its respective standard deviation, instead of the median.

3) Regarding the use of "concomitant agents" described in table 1, were the patients not using "Cevimeline/Pilocarpine and Topical clonazepan" during baseline data collection? If the answer is yes, I suggest removing them from the table.

4)I would like to know if the term "second visit" described in table 2 corresponds to the term "first follow-up visit" as described in the results and discussion of the manuscript. I suggest standardizing the term so as not to confuse the reader. The same applies to the term: "second follow-up visit" which in table 2 is described as third visit.

5) In table 2 - item: Other oral symptoms, I think the authors forgot to add an asterisk after the word "other", to refer to the note described below the table.

6) The authors succinctly described the limitations of the study at the end of the discussion, in which they reported that the period of patient follow-up was variable, ranging from 29 days to 195 days, making it difficult to compare patients' response to treatment at similar follow-up intervals. Therefore, there is doubt regarding the statistical analysis for the variable referring to the sensation of pain/burning mouth, as the authors mention "NRS score (p < 0.01)". If this analysis is truly viable, I missed a specific table showing the statistical significance ("p" value) in relation to the analysis of scores for mouth pain/burning.

Note: Normally, tables presented in manuscripts do not have side borders or internal borders.

6. PLOS authors have the option to publish the peer review history of their article (what does this mean?). If published, this will include your full peer review and any attached files.

Reviewer #1: No

Reviewer #2: No

---

## [Author Response · Author response to Decision Letter 0]

12 Oct 2023

10/11/2023

Dr. Claudia Sommer

Academic Editor

Plos One

Author’s Response Letter for Manuscript ID: [PONE-D-23-22592]

Title: Topical gabapentin solution for the management of Burning Mouth Syndrome: a retrospective study

Dear Dr. Claudia Sommer

We are thankful for the reviewers’ thoughtful comments. Our point-by-point individual responses to comments by the reviewers are addressed below.

Please find enclosed the revised manuscript with “track changes”, and please, let me know if you have any questions.

Sincerely,

Amanda Gramacy, DDS

Alessandro Villa, DDS, PhD, MPH

University of California, San Francisco School of Dentistry

Comments from the Editors and Reviewers:

Reviewer #1: The Authors presented an interesting clinical retrospective study on topical solution of gabapentin for BMS. The subject is clinically important due to limited effectiveness of available treatment modalities.

The manuscript needs some minor revisions before publication.

Line 59-60 - propose to describe the pathophysiology of BMS more precisely (alternatively in discussion) , including some molecular and cellular mechanisms contributing to neuropathic component of BMS, what is important regarding the mechanism of action of gabapentin in this clinical entity. In the pathophysiology of BMS , the nociplastic component may be involved, propose to add .

Response: We thank the reviewer for the comment. We have added a section in the introduction to address and more concisely describe the pathophysiology of BMS. It now reads as: 

“The pathophysiology of BMS is poorly understood and may be related to both physiological and psychological components [5]. Evidence suggests that the mechanism of BMS is neuropathic in nature [5]. In the neuropathic pathophysiology theory, sensory dysfunction is associated with small and/or large fiber neuropathy where there is axonal degeneration of epithelial and subpapillary nerve fibers in the affected epithelium of the oral mucosa [6]. There is also an abnormal interaction between the sensory functions of facial and trigeminal nerves. BMS is considered a nociplastic pain disorder in that the pain arises from altered nociception despite no evidence of tissue damage causing the activation of peripheral nociceptors and no evidence of a lesion causing the pain [7].”

section Discussion :

propose to discuss more precisely the molecular and cellular mechanism of action of gabapentin , particularly in terms of topical administration. Which mechanisms of gabapentin MOC might be involved in BMS treatment , which types of cells may be targeted by gabapentin? Propose to refer as well to clinical studies presenting other pain syndromes treated with topical gabapentin, including mucositis if there are studies available.

Response: We thank the reviewer for the comment. We have added a section in the discussion that reviews the current literature of the mechanism of action of gabapentin both systemically and topically. It now reads as:

The precise mechanism of action of gabapentin remains unclear as to how it exerts its therapeutic effect. Systemic gabapentin inhibits the action of α2δ-1 subunits, thus decreasing the density of pre-synaptic voltage-gated calcium channels and subsequent release of excitatory neurotransmitters [7]. A topical gabapentin approach has shown to have a positive analgesic effect with fewer side effects in other neuropathic pain syndromes such as severe postherpetic neuralgia and chronic sciatic nerve constriction injury [22, 23]. We can hypothesize that the mechanism of action of topical gabapentin is similar to systemic gabapentin by blocking the α2δ-1 subunits present in nociceptive neurons, but that the analgesic effect is produced locally. Nociceptors in mucosal barriers contain various types of receptors that bind different ligands which influence the generation of pain transmitting action potentials. Topical gabapentin could traverse tissue and increase the nociceptive threshold by stabilizing the membranes of specific nociceptors [24].

Despite small group of patients studied , the study is relevant for clinical purposes and after minor revisions may be published

Reviewer #2: - 

Introduction

1) On line 55, I suggest removing the sentence: “tingling or changes in salivary gland function”.

We thank the reviewer for the comment. We have removed this phrase from the sentence. It now reads as:

“The primary symptoms include oral burning although patients may present with other sensory symptoms including xerostomia and dysgeusia [3].

- Materials and Methods

1) I suggest adding to your methodology the inclusion of patients with primary and secondary BMS, since it was mentioned that all patients in the sample had comorbidities such as type II diabetes, hypothyroidism,...

We thank the reviewer for the comment. In this study, we included patients with both primary and secondary BMS. We have updated the methodology section to clarify this. It now reads as: 

“We included patients that had both primary and secondary BMS. Patients with an underlying medical condition were stable and have had these medical conditions for many years.”

- Results

1)I think there is no need to subdivide the results by topic.

We thank the reviewer for the comment. We agree and we have taken out the subtopics in the results section.

2) I suggest removing the information regarding the patients' age from table 1, since it has already been mentioned in the text. It is preferable to use the mean and its respective standard deviation, instead of the median.

We thank the reviewer for the comment. We have removed the information regarding the patients’ age from Table 1 and have added mean and its respective standard deviation.

3) Regarding the use of "concomitant agents" described in table 1, were the patients not using "Cevimeline/Pilocarpine and Topical clonazepan" during baseline data collection? If the answer is yes, I suggest removing them from the table.

We thank the reviewer for the comment. The patients were indeed not using Cevimeline/Pilocarpine and topical clonazepam during baseline data collection. This information has been removed from the table.

4)I would like to know if the term "second visit" described in table 2 corresponds to the term "first follow-up visit" as described in the results and discussion of the manuscript. I suggest standardizing the term so as not to confuse the reader. The same applies to the term: "second follow-up visit" which in table 2 is described as third visit.

We thank the reviewer for the comment. We have streamlined the terms. A first follow up is now called “second visit” and a second follow up is now called a “third visit”

5) In table 2 - item: Other oral symptoms, I think the authors forgot to add an asterisk after the word "other", to refer to the note described below the table.

We thank the reviewer. We have added the asterisk to Table 3 (previously Table 2).

6) The authors succinctly described the limitations of the study at the end of the discussion, in which they reported that the period of patient follow-up was variable, ranging from 29 days to 195 days, making it difficult to compare patients' response to treatment at similar follow-up intervals. Therefore, there is doubt regarding the statistical analysis for the variable referring to the sensation of pain/burning mouth, as the authors mention "NRS score (p < 0.01)". If this analysis is truly viable, I missed a specific table showing the statistical significance ("p" value) in relation to the analysis of scores for mouth pain/burning.

We thank the reviewer for the comment. We have added a new table that reflects this calculation. Please see Table 1.

Note: Normally, tables presented in manuscripts do not have side borders or internal borders.

We thank the reviewer for the comment. We have removed the side and internal borders from the tables.

---

## [Decision Letter · Decision Letter 1]

27 Nov 2023

Topical gabapentin solution for the management of Burning Mouth Syndrome: a retrospective study

PONE-D-23-22592R1

Dear Dr. Villa,

We’re pleased to inform you that your manuscript has been judged scientifically suitable for publication and will be formally accepted for publication once it meets all outstanding technical requirements.

Kind regards,

Claudia Sommer

Academic Editor

PLOS ONE

Additional Editor Comments (optional):

Reviewers' comments:

Reviewer's Responses to Questions

**Comments to the Author**

1. If the authors have adequately addressed your comments raised in a previous round of review and you feel that this manuscript is now acceptable for publication, you may indicate that here to bypass the “Comments to the Author” section, enter your conflict of interest statement in the “Confidential to Editor” section, and submit your "Accept" recommendation.

Reviewer #2: All comments have been addressed

2. Is the manuscript technically sound, and do the data support the conclusions?

Reviewer #2: Yes

3. Has the statistical analysis been performed appropriately and rigorously? 

Reviewer #2: Yes

4. Have the authors made all data underlying the findings in their manuscript fully available?

Reviewer #2: Yes

5. Is the manuscript presented in an intelligible fashion and written in standard English?

Reviewer #2: Yes

6. Review Comments to the Author

Reviewer #2: (No Response)

7. PLOS authors have the option to publish the peer review history of their article (what does this mean?). If published, this will include your full peer review and any attached files.

Reviewer #2: No

---

## [Editor Report · Acceptance letter]

4 Dec 2023

PONE-D-23-22592R1 

Topical gabapentin solution for the management of Burning Mouth Syndrome: a
retrospective study 

Dear Dr. Villa:

I'm pleased to inform you that your manuscript has been deemed suitable for publication in PLOS ONE. Congratulations! Your manuscript is now with our production department. 

Kind regards, 

on behalf of

Prof. Dr. Claudia Sommer 

Academic Editor

PLOS ONE